# Relationship between Phenotypic and Genotypic Resistance of Subgingival Biofilm Samples in Patients with Periodontitis

**DOI:** 10.3390/antibiotics12010068

**Published:** 2022-12-30

**Authors:** Moritz Sparbrod, Yann Gager, Anne-Katrin Koehler, Holger Jentsch, Catalina-Suzana Stingu

**Affiliations:** 1Institute for Medical Microbiology and Virology, University Hospital Leipzig, 04103 Leipzig, Germany; 2Parox GmbH, 04103 Leipzig, Germany; 3Center of Periodontology, University Hospital Leipzig, 04103 Leipzig, Germany

**Keywords:** resistance genes, subgingival biofilm, phenotypic resistance

## Abstract

The phenotypic expression of antibiotic resistance genes (ARGs) can hamper the use of antibiotics as adjuncts to subgingival instrumentation in the treatment of periodontitis patients. The aim of the study was to analyze the relationship between the phenotypic and genotypic resistance against ampicillin-sulbactam, clindamycin, doxycycline and metronidazole of subgingival biofilm samples from 19 periodontitis patients. Samples were analyzed with shotgun sequencing and cultivated anaerobically for 7 days on microbiological culture media incorporating antibiotics. All growing isolates were identified to the species level using MALDI-TOF-MS and sequence analysis of the 16S ribosomal RNA (rRNA) gene. Phenotypic resistance was determined using EUCAST-breakpoints. The genetic profile of eight patients matched completely with phenotypical resistance to the tested antibiotics. The positive predictive values varied from 1.00 for clindamycin to 0.57 for doxycycline and 0.25 for ampicillin-sulbactam. No sample contained the *nimI* gene. It can be concluded that antibiotic resistance may be polygenetic and genes may be silent. Every biofilm sample harboring *erm* genes was phenotypic resistant. The absence of *cfx* and *tet* genes correlated to 100%, respectively, to 75%, with the absence of phenotypic resistance. The absence of *nimI* genes leads to the assumption that constitutive resistance among several species could explain the resistance to metronidazole.

## 1. Introduction

Subgingival instrumentation is part of the stepwise therapy of periodontitis and usually conducted with hand- and/or power-driven (e.g., sonic/ultrasonic devices) instruments. It aims to remove the dysbiotic subgingival biofilm that consists predominantly of streptococci and *Actinomyces* species as primary and obligate anaerobic bacteria (such as Bacteroidaceae species and spirochaetes) as secondary colonizers [1,2,3]. As a result, it paves the way for a healthy subgingival microflora that is essential in terms of soft tissue inflammation status. The therapeutic effectiveness has been proven in a variety of studies [4,5,6]. However, numerous studies reported clinical benefits in the adjunctive administration of systemic antibiotics in recent years [7,8,9,10]. Commonly used are β-lactam antibiotics and metronidazole. In case of intolerance to metronidazole, clindamycin is the antibiotic of choice. Tetracyclines (e.g., doxycycline) are systemically administered as well [11] and it was further found that the local application may lead to clinical improvements [12]. Therefore, it seems indicated to take advantage of this adjuvant therapy in risk groups, such as young adults with severe periodontitis, patients with inaccessible points or persistent pockets. Nevertheless, there is lack of evidence in long-term data (>12 months) to support clinical protocols of which products should be used in which dosages. Generally, excess use of antibiotics should be carefully considered due to justified doubts regarding the overuse and the development of antibiotic resistance. In recent time, numerous studies reported increased resistance to several antibiotics, especially in anaerobic bacteria [13,14]. One of the main concerns in antibiotic resistance is the presence of antibiotic resistance genes (ARGs) in different environments and their horizontal gene transfer, especially in a biofilm environment [15]. The ARGs *erm* were observed as one possible resistance mechanism against clindamycin and showed high prevalence in the subgingival biofilm [16]. *Tet* genes which were identified as being responsible for resistance to doxycycline were frequently detected as well [17]. β-lactam antibiotic resistance is mainly linked to the presence of *cfxA* genes. *CfxA* is not merely highly prevalent in β-lactamase producing oral anaerobic bacteria but also in the oral biofilm of periodontitis patients [18,19]. Metronidazole resistance is mainly linked to the presence of *nim* genes which encode for nitro-imidazole-reductases [20]. However, few data are available about the relationship between the prevalence of ARGs and phenotypic resistance of biofilm samples in patients with periodontitis. Previous studies solely investigated the phenotypic resistance to antibiotics in the subgingival microflora [21], analyzed antimicrobial resistance in periodontal abscesses of obligate anaerobes [22] or focused on certain species (*Aggregatibacter actinomycetemcomitans*, *Porphyromonas gingivalis*, *Prevotella intermedia*, *Fusobacterium nucleatum* and *Parvimonas micra*) [23,24]. By using modern analytical methods, such as shotgun metagenomic sequencing and MALDI-TOF-MS, we were able to evaluate the bacterial diversity and detect ARGs, even when they occurred in low abundance. Therefore, the aim of the study was to determine the relationship between the phenotypic and genotypic resistance against ampicillin-sulbactam, clindamycin, doxycycline and metronidazole of subgingival biofilm samples in periodontitis patients, using conventional methods.

## 2. Materials and Methods

### 2.1. Patients

A total of 19 patients with periodontitis were enrolled in this study. The research was conducted from August 2020 to July 2022. Diagnosis of periodontitis was based on clinical and radiographic findings prior to the study. We respected the criteria of the novel S3 level clinical practice guideline for the treatment of stage I–III periodontitis by the European Federation of Periodontology [25]. The clinical measurements including probing depth (PD), clinical attachment loss (CAL) and bleeding on probing (BOP) were recorded in a six-point measurement per tooth (mesiobuccal, buccal, distobuccal, mesiooral, oral and distooral) using a periodontal probe (PCP-UNC 15, Hu-Friedy Manufacturing Co., Chicago, IL, USA). PD is defined as the distance from the base of the pocket to the most coronal point on the gingival margin, measured in mm. CAL is, in the case of periodontitis, the distance between the cemento-enamel-junction and the base of the pocket, measured in mm [26]. BOP is induced while measuring PD and CAL due to manipulation of the tissue in the periodontal pocket and is given as present (positive) or absent (negative) for the tested site. Inclusion criteria were a residual dentition of at least 16 teeth, presence of supra- and subgingival plaque and a PD of ≥3 mm. Inflammatory diseases were no reason for exclusion, as one of the patients suffered with type 2 diabetes. Criteria for being excluded from the study were an age of under 18, antimicrobial therapy 3 months prior to the study, pregnancy and breastfeeding period, recently received professional supragingival plaque removal and/or subgingival instrumentation within the last 12 months.

### 2.2. Sampling Process

Samples of the subgingival biofilm were taken with four paper points (ISO 50, Roeko GmbH, Langenau, Germany) per patient. Each paper point was used for one appropriate pocket per quadrant with a depth of 3–10 mm. They were left in the pocket for 30 s and then pooled in 2 mL brain-heart infusion broth (Oxoid, Wesel, Germany). The samples were immediately carried to the laboratory and vortexed (IKA-Labortechnik, Staufen, Germany) for 60 s at 2.000 rpm; 1 mL of the broth was used for microbiological culture. The remaining broth was frozen at −80 °C and later used for next generation sequencing (NGS) via shotgun.

### 2.3. Microbiological Growth and Identification of Isolates

Two series of Columbia agar plates (Oxoid, Basingstone, UK, supplemented with 5% sheep blood, 5 μg/L hemin, 1 μg/L Vit. K1) were prepared by incorporating ampicillin-sulbactam, clindamycin, doxycycline and metronidazole in breakpoint concentrations shown in Table 1. For each antibiotic in each concentration, a separate plate was made. A combination disk test was performed using a 5 µg vancomycin disk and a 10 µg colistin sulphate disk to provide a better differentiation between Gram-postive and Gram-negative bacteria [27]. The plates were inoculated with 100 μL of the samples and then incubated in an anaerobic station (Whitley MG 1000 anaerobic workstation, Meintrup DWS Laborgeräte GmbH, Lähden-Holte, Germany) at 37 °C for 7 days. The following species were used as control: *Bacteroides ovatus*, *Bacteroides fragilis* and *Bacteroides thetaiotaomicron*. Further, two Columbia agar plates, without incorporated antibiotics, were used as a growth control. The control plates were incubated for 48 h in the same anaerobic station. Phenotypic morphology (including the hemolytic features, colony size, pigmentation edge situation and transparency) was used to differentiate the strains. All the grown colonies that were presumptive assigned to a specific bacterial species were subjected to identification using VITEK-MS (bioMeriuex, Lyon, France) on the V2.0 Knowledge Base for clinical use. Polymeric target slides (bioMeriuex) were prepared by picking the colonies with an inoculation loop and applying them onto the spots in duplicate. Following the manufacturer’s manual, *Escherichia coli* was then spotted onto the slides to calibrate and control the method. After air-drying for 5 min, we covered the spots with 1 μL matrix solution (VITEK MS CHCA, α-cyano-4hydroxycinnamic acid). Subsequently, the targets were air-dried for a further 10 min and then loaded into the mass spectrometer. The spectra were compared to the VITEK MS V2.0 Knowledge Base for clinical use. For the unidentified strains, we repeated the protocol twice. Successful identification was considered at 99.9% probability. All remaining unidentified strains were subjected to the sequence analysis of the 16S ribosomal RNA (rRNA) gene [28]. In the first step, we extracted the DNA using MagNA Pure 96 (Roche Diagnostics, Mannheim, Germany), following the manufacturer’s instructions. A PCR-MasterMix with a total volume of 50 μL reaction mixture, containing a final concentration of 50 mM KCl, 10 mM Tris-hydrochloride (pH 8.3), 1.5 mM MgCl_2_, 0.2 mM of each desoxynucleoside triphosphate, 0.4 μM of each primer (TIB Molbiol, Berlin, Germany) [29], 1 U of Taq DNA polymerase (AmpliTaq, Thermo Fisher Scientific, Waltham, MA, USA) and 2 μL bacterial DNA, was prepared. An overview of the used primers is shown in Table 2. PCR was performed by thermocycler TGradient96 and TPersonal (Biometra, Göttingen, Germany). The procedure was: 95 °C for 2 min, then 35 cycles of 94 °C for 30 s, 52 °C for 40 s, 72 °C for 50 s and a further 72 °C for 8 min. A positive and negative control were used in each run. PCR products were analyzed in a 1.5% agarose gel, stained with ethidium bromide and visualized under UV light. Amplified products were purified by MSB Spin PCRapace Kit (Stratec Molecular, Berlin, Germany). A total volume of 10 μL reaction mixture contained 4 μL of Big-Dye v. 3.1 (Thermo Fisher Scientific, Waltham, MA, USA), 1 μM of forward/reverse primer and 2 μL of cleaned PCR product. PCR was run by thermocycler once more in the following sequence: 95 °C for 2 min, 30 cycles of 95 °C for 30 s, 52 °C for 15 s and 60 °C for 4 min. In the last step, PCR products were purified by DyeEx 2.0 Spin Kit and dried with SpeedVac DNA130 (Thermo Fisher Scientific). The amplified products were sequenced with an ABI 3130xl Genetic Analyzer (Applied Biosystems, Foster City, CA, USA). The software Geneious Prime 2021.2.2 was used to match the single stranded amplified products. The comparison was performed using the Standard Nucleotide BLAST function through the NCBI BLAST web service. The isolates had a minimum of 99% identity, except for one isolate of the *Bacteroides ovatus* like-species and one isolate of the *Anaplasma ovis* like-species. These two had a 96% identity.

### 2.4. ARG Testing

All biofilm samples were screened for 3.123 resistance genes from the ResFinder Database including the following ARGs: *cfxA*, *cfxA3*, *cfxA4*, *cfxA5*, *ermB*, *ermF*, *ermG*, *mefA*, *nimI*, *tet32*, *tetM*, *tetO* and *tetQ*. NGS testing was performed via shotgun. The process consisted of elution, extraction, microbiome DNA enrichment, ultrasound fragmentation, library preparation, pooling and sequencing. Quality controls for the different steps were performed using TapeStation 2200 (Agilent, Santa Clara, CA, USA). DNA was extracted with the QIAmp PowerFecal Pro DNA Kit (QIAGEN, Hilden, Germany) following the manufacturer’s instructions with some modifications. Instead of using 250 mg of stool for step 1, we centrifuged 1 mL brain-heart infusion broth (Oxoid, Wesel, Germany) for 20 min by 13,000 rpm, removed 750 μL from the supernatant and continued the protocol with the remaining 250 μL. For the homogenization of samples (step 2), we used the option C with the Tissue Lyser LT (QIAGEN) for 10 min with a speed of 25 1/s. The following centrifugation (step 3) was performed at a speed of 13,000 rpm for 1 min. We transferred 600 μL of the supernatant in a rotor adapter (step 4). The steps 5 to 17 were performed on the QIAcube using the program “DNA_PowerSoilPro_Soil_IRT_V1”. Samples were enriched for microbial DNA using the NEBNext Microbiome DNA Enrichment kit (New England BioLabs, Ipswich, MA, USA). Samples enriched for microbial DNA were then fragmented with the M220 Focused-ultrasonicator (COVARIS). DNA fragments were then prepared using the TruQuant DNA Library Preparation kit (GenXPro, Frankfurt). Libraries were sequenced at 2 × 151 cycles with a High Output Flow Cell Cartridge v.2.5 on an Illumina NextSeq 500 (Illumina, San Diego, CA, USA).

### 2.5. Bioinformatics

The company GenXPro used bcl2fastq (Illumina, San Diego, CA, USA) for demultiplexing and Cutadapt (Dortmund, Germany) [30] for pre-processing. The software CLC Genomics Workbench 20.0.03 (Qiagen, Aarhus, Denmark) was used for the rest of the bioinformatic analyses. The function “De Novo Assembly” with default parameters (except for word size 23) was used to obtain a list of contigs from trimmed reads. We obtained 15,732,604 to 30,961,594 paired reads per sample (sample mean = 23,992,537). After trimming and de novo assembly, we maintained 238,126 to 1,046,320 contigs per sample (sample mean = 725,981). The function “Find Resistance” from the Microbial Genomics Module 3.0 was used to compare the trimmed reads and contigs with the database Find Resistance (accessed on 28 January 2021) with 98% minimum identity and 60% minimum length. ARGs found in contigs were considered as strong evidence. A number of >10 reads was defined as medium evidence (discrete signal) and a number of 1–10 reads as low evidence (discrete signal or artefact) for ARGs.

### 2.6. Predictive Values

The connection of phenotypic resistance and detected ARGs in the biofilm samples was analyzed using the standard mathematic formulas for predictive values. A true-positive test result was defined as presence of resistance genes in the sample and finding of phenotypic resistance in growing strains. A false-positive result was defined as prevalence of resistance genes in the samples and phenotypic susceptibility in growing strains. The criterion used for a true-negative result was absence of ARGs. In case of phenotypic resistance but absence of ARGs, we assigned a false-negative test result.

### 2.7. Comparison of Shotgun Analysis and MALDI-TOF-MS/16S Sequence Analysis

We manually selected the contigs with ARGs for further analysis (e.g., contig x from patient y with ARG z). The contigs were compared with all the sequences listed in GenBank using the Standard Nucleotide BLAST function through the NCBI BLAST web service with default parameters from CLC Genomics Workbench. In this way, we detected both the ARGs and the species of bacteria in the samples. We then compared the BLAST output with the microbiological results to check if we could identify the same species with shotgun analysis and MALDI-TOF-MS/sequence analysis of the 16S ribosomal RNA (rRNA) gene.

## 3. Results

### 3.1. Isolated Bacterial Strains

A total of 63 bacterial strains were isolated from the 19 subgingival biofilm samples of the patients. The demographic data and clinical parameters of the participants are shown in Table 3. The results of identification via MALDI-TOF-MS/16S sequencing are summarized in Table 4; 45 (71.43%) isolates were obligate anaerobic bacteria and 16 strains (25.40%) belonged to the *Streptococcus* genus. The most anaerobic bacteria (57.78%) were Gram-negative. *Prevotella* (11 isolates) was the most frequently detected genus, followed by *Capnocytophaga* (nine isolates), *Gemella* (six isolates) and *Actinomyces* (five isolates). Other genera included *Bacteroides*, *Anaplasma*, *Cutibacterium*, *Solobacterium*, *Alloscardovia*, *Microbacterium*, *Bulleida*, *Eggerthia* and *Fusobacterium*. Among the six isolated streptococcal species, *S. mitis/oralis* (five isolates) and *S. constellatus* (four isolates) were the most prevalent. *S. anginosus* was found in three samples. The remaining two strains belonged to the genera *Corynebacterium* and *Bacillus*.

### 3.2. Prevalence of ARGs

The prevalence of resistance genes are shown in Table 5. Resistance genes for ampicillin-sulbactam were identified in four samples (21.05%). All the samples harbored the *cfxA3* gene. The genes *cfxA3*, *cfxA3*4 and *cfxA3*5 appeared always together. The ARGs for clindamycin occurred in eight samples (42.11%). *ErmF* (five samples) was the most frequently detected gene in this group. *MefA*, *ermB* and *ermG* genes were identified in three, two and one sample(s), respectively. Seven samples (36.84%) harbored ARGs for doxycycline. *Tet32* (four samples) was the resistance gene with the highest prevalence in this group, followed by *tetM* (three samples), *tetQ* (two samples) and *tetO* (one sample). The resistance gene nimI, responsible for metronidazole resistance, could not be detected in any of the samples.

### 3.3. Predictive Values of Resistance Genes

In Table 6, the relationship of detected ARGs and phenotypic resistance for each biofilm sample can be seen. Corresponding genes and antibiotic resistance phenotypes were highlighted in green. Resistance against ampicillin-sulbactam was found in one of the four samples that harbored *cfxA*, *cfxA3*, *cfxA4*, or *cfxA5* genes. The positive predictive value (PPV) was 0.25. All the 15 samples without resistance genes for ampicillin-sulbactam were phenotypic susceptible, consequently, the negative predictive value (NPV) was 1.00. None of the samples with clindamycin-associated ARGs were susceptible and eight were resistant. Among the samples without resistance genes, five were phenotypic resistant and six were sensitive. Consequently, a PPV of 1.00 and an NPV of 0.55 were determined. Resistance against doxycycline was identified in four of the seven samples with prevalence of ARGs. The PPV was 0.57; 12 samples did not contain resistance genes for this antimicrobial agent. The NPV was 0.75 with nine phenotypic susceptible and three resistant samples. It was unfeasible to calculate the PPV for metronidazole, because no *nimI* gene was detected. All the samples were susceptible. The NPV was 1.00.

### 3.4. Identification via Shotgun Analysis

The comparison of the different methods used for identification can be seen in Table 7. We only illustrated these outcomes that were equal on the genus or species level. Matching results on the species level were detected in biofilm sample 1 and 5. Given the fact that the species of the isolates are equal to species that were found on the contigs, we can validate the phenotypic expression of the ARGs. Both bacterial isolates were resistant against clindamycin. Sample 1 harbored resistant *Prevotella intermedia* and sample 5 resistant *Bacteroides ovatus* species. In a further three samples, we obtained the same genus. These samples were all clindamycin resistant as well. Samples 2 and 8 contained *Provotellae* and sample 7 resistant *Gemellae*.

## 4. Discussion

In the present study, subgingival biofilm samples of 19 periodontitis patients were investigated regarding their phenotypic and genotypic resistance. A wide range of studies about antibiotic resistance in subgingival bacteria has been performed, but few data are given about the association between resistance genes and phenotypic resistance [13,14,15,16,17,19,20,21,22,23,24,31,32,33,34,35].

Overall, the genetic profile of eight from 19 samples matched completely with phenotypical resistance to the tested antibiotics. In each of the 11 remaining samples, a minimum of low evidence reads were detected for ARGs, if phenotypic resistance was observed. Therefore, all the 19 samples were included for determination of the predictive values.

In agreement with a previous study, streptococci were the most frequently detected species with 16 strains [17]. Moreover, *Streptococcus mitis*/*oralis* were also the species with the highest prevalence in this group [17]. *Prevotella* was the second most detected genus with 11 isolates. By comparing the different methods that were used for identification, we observed that the most resistant isolates were *Prevotellae*, which corresponds with former studies that described high prevalence of bacterial strains of this genus in the subgingival microflora of periodontitis patients and periodontal abscesses as well [17,22].

We investigated the prevalence of *cfxA*, *cfxA3*, *cfxA4* and *cfxA5* genes in this study. The presence of these genes is responsible for resistance to penicillins and cephalosporins. The most important factor for resistance is the expression of β-lactamase, for which the *cfxA* genes are coding [18,36]. Four biofilm samples harbored this type of gene, all of them being positive for *cfxA3* and three of them for all the tested ARGs. A PPV of 0.25 was calculated. Samples that were positive tested for the *cfxA3* gene and phenotypic resistant contained isolates of the *Streptococcus mitis*/*oralis* and *Fusobacterium nucleatum* species. These findings contrast with a former study where 42% of the isolates containing *cfxA* genes were resistant to ampicillin [18]. In another study, 53% of the strains containing ARGs (all belonging to the *Prevotella* genus) were resistant to β-lactam-antibiotics [22].

Resistance to clindamycin can be caused by the presence of *ermB*, *ermf*, *ermG* and *mefA*. There are two major mechanisms for resistance. First, *erm* genes are encoding for a methylase which methylates the ribosomal target of the antibiotic [31]. Second, *mefA* encodes an effective efflux pump [32]. These ARGs had the highest PPV in this study with 1.00. All the eight samples that harbored at least one of these genes were phenotypically resistant. Five (62.5%) of these samples contained *ermF* which matches with a former study that detected this gene in 60.7% of clindamycin- and/or roxithromycin-resistant *Prevotella* species [22]. In the present study, *erm* genes were, with an occurrence in eight samples, the most detected resistance genes. This result confirms the findings of another study where *erm* has been reported as the ARGs with the highest prevalence in saliva and subgingival plaque samples as well [16]. It was found out that *erm* is often harbored on mobile genetic elements like conjugative transposons from the *Tn916* family [33,34,37]. It has been shown that Viridans streptococci and *Gemella* species take use of conjugative transfer to spread resistance genes to major streptococcal pathogens like *Streptococcus pneumoniae* and *Streptococcus pyogenes* [34]. Phenotypic resistant samples in this study contained the following genus: *Prevotella*, *Bacteroides*, *Anaplasma*, *Corynebacterium*, *Capnocytophaga* and *Streptococcus*. By comparing the outputs of the two different methods used for identification (MALDI-TOF-MS/16S sequencing and shotgun analysis of the contigs), we were able to verify resistant *Prevotellae*, *Bacteroides* and *Gemellae*. It can be assumed that these isolates showed phenotypic expression of their ARGs.

A PPV of 0.57 was investigated for *tet32*, *tetM*, *tetO* and *tetQ*. None of the *tetQ* positive samples was phenotypic resistant. That contrasts with another study which detected *tetQ* in 50% of doxycycline-resistant strains [22]. Resistance to doxycycline is associated with these genes due to ribosomal protection [33]. In the present study, *tet* genes had the second highest prevalence among the ARGs. Seven samples harbored at least one of these resistance genes, but three of them were susceptible to doxycycline. *Tet32* was the most frequently detected gene of this group with an occurrence in four samples, followed by *tetM*. A former study discovered *tetM* most often [16]. All the three subgingival biofilm samples that were tested positive for *tetM* showed phenotypic resistance. In the present study, the coincident occurrence of *tet* genes and *erm/mef* genes was observed in four samples. Similar findings were researched in a previous study that identified a strong association in the presence of *tetM* and *ermB* [38]. Both samples containing *ermB* in this study harbored *tetM*. As previously mentioned for *erm* genes, *tet* genes have also been described to be transferred on conjugative transposons of the *Tn916* family and disseminated by oral streptococci and anaerobic bacterial species [33,37].

None of the samples contained the *nimI* gene which is responsible for metronidazole resistance. Therefore, the PPV could not be calculated. The absence of *nimI* corresponds with the findings of other studies which also could not detect *nim* genes from samples of the subgingival flora [16,39]. Even though no ARGs were identified, 16 samples were phenotypically resistant against metronidazole due to constitutive resistance in *Streptococcus*, *Gemella*, *Actinomyces*, *Capnocytophaga*, *Cutibacterium* and *Alloscardovia* species [35,40,41,42].

There are some limitations in the present study. The software could not assemble contigs from the biofilm samples in every case, due to low DNA concentrations. For this reason, we included samples with reads (1–10 overlapping sequences) for ARGs to verify the presence of ARGs and further compare them to their phenotypic resistance. Moreover, there is lack of information about ARGs that are carried by non-cultivable species.

## 5. Conclusions

In summary, within the limits of this study, it can be concluded that antibiotic resistance may be polygenetic and genes may be silent. It was shown that the predictive values for phenotypic resistance vary. However, it was found that every biofilm sample harboring *erm* genes was phenotypic resistant. In addition, the absence of *cfx* and *tet* genes correlated to 100%, respectively to 75%, with the absence of phenotypic resistance. The absence of *nimI* genes leads to the assumption that constitutive resistance among several species could explain the resistance to metronidazole. Further studies are needed to investigate resistance mechanism when ARGs are silent.

## Figures and Tables

**Table 1 antibiotics-12-00068-t001:** Antibiotic plate concentration.

Antibiotics	Concentration in Agar Plates (mg/L)
Ampicillin-Sulbactam	2; 4 and 8
Clindamycin	2 and 4
Doxycycline	2 and 4
Metronidazole	2 and 4

**Table 2 antibiotics-12-00068-t002:** Primers used in this study.

Name	Sequence 5′–3′	Reference
BAK	AGT TTG ATC HTG GCT CAG	[29]
PC 3mod	GGA CTA CHA GGG TAT CTA AT	[29]

**Table 3 antibiotics-12-00068-t003:** Demographic data and clinical parameters of the participants of the study.

Parameter	Data (Mean ± SD, n = 19)
Age (years)	56.26 ± 13.45
Sex (male/female, %)	47/53
Smoker (%)	32
PD (sites) *	6.16 ± 1.73
CAL (sites) *	5.63 ± 2.89
BOP (positive/negative, sites, %) *	58/42

* Mean of the six sampling sites.

**Table 4 antibiotics-12-00068-t004:** Identification of isolated bacteria from samples.

Taxonomic Rank		n	%
Genus	Species		
*Streptococcus*	*S. mitis/oralis*	5	26.32
	*S. constellatus*	4	21.05
	*S. anginosus*	3	15.79
	*S. intermedius*	2	10.53
	*S. massiliensis*	1	5.26
	*S. gordonii*	1	5.26
*Prevotella*	*P. maculosa*	4	21.05
	*P. nigrescens*	2	10.53
	*P. veroralis*	1	5.26
	*P. salivae*	1	5.26
	*P. nanceiensis*	1	5.26
	*P. intermedia*	1	5.26
	*P. buccae*	1	5.26
*Capnocytophaga*	*C. gingivalis*	6	31.58
	*C. ochracea*	2	10.53
	*C. granulosa*	1	5.26
*Gemella*	*G. morbillorum*	6	31.58
*Actinomyces*	*A. meyeri*	3	15.79
	*A. odontolyticus*	2	10.53
*Bacteroides*	*B. thetaiotaomicron*	1	5.26
	*B. pyogenes*	1	5.26
	*B. ovatus/xylanisolvens*	1	5.26
*Anaplasma*	*A. ovis*	1	5.26
	*A. phagocytophilum*	1	5.26
*Cutibacterium*	*C. acnes*	2	10.53
*Solobacterium*	*S. moorei*	2	10.53
*Alloscardovia*	*A. omnicolens*	1	5.26
*Corynebacterium*	*C. durum*	1	5.26
*Microbacterium*	*M. flavescens/laevaniformans*	1	5.26
*Bulleida*	*B. extructa*	1	5.26
*Eggerthia*	*E. catenaformis*	1	5.26
*Bacillus*	*B. cereus group*	1	5.26
*Fusobacterium*	*F. nucleatum*	1	5.26

n, Number of biofilm samples that harbored the species.

**Table 5 antibiotics-12-00068-t005:** Prevalence of ARGs in the samples.

Antibiotic	Resistance Genes	Number of Biofilm Samples
Ampicillin-Sulbactam	*cfxA*	3
	*cfxA3*	4
	*cfxA4*	3
	*cfxA5*	3
Clindamycin	*ermB*	2
	*ermF*	5
	*ermG*	1
	*mefA*	3
Doxycycline	*tet32*	4
	*tetM*	3
	*tetO*	1
	*tetQ*	2
Metronidazole	*nimI*	0

**Table 6 antibiotics-12-00068-t006:** Connection of resistance genes and phenotypic resistance.

Sample	Resistance Genes				Phenotypic Resistance of the Samples								
	Ampicillin-Sulbactam	Clindamycin	Doxycycline	Metronidazole	Ampicillin-Sulbactam			Clindamycin		Doxycycline		Metronidazole	
					2 mg/L (S)	4 mg/L (S)	8 mg/L (R)	2 mg/L (S)	4 mg/L (R)	2 mg/L	4 mg/L	2 mg/L (S)	4 mg/L (R)
1	0	*ermF*	0	0	ø	ø	ø	+	+	ø	ø	+	ø
2	*cfxA*, *3*, *4*, *5*	*ermF*	0	0	ø	ø	ø	ø	+	ø	ø	+	+
3	0	0	0	0	ø	ø	ø	+	+	+	+	+	+
4	0	0	0	0	ø	ø	ø	ø	+	ø	ø	+	+
5	*cfxA3*	*erm(B*, *F*, *G)*, *mefA*	*tet32*, *tetM*	0	+	+	+	+	+	+	+	+	+
6	0	0	0	0	ø	ø	ø	ø	ø	ø	ø	+	+
7	0	*mefA*	0	0	ø	ø	ø	+	+	+	ø	+	+
8	0	*ermF*	0	0	ø	ø	ø	+	+	ø	ø	+	+
9	0	*ermF*	*tet32*	0	ø	ø	ø	+	+	ø	ø	+	+
10	0	0	0	0	ø	ø	ø	+	+	ø	ø	+	+
11	0	*mefA*	*tet32*	0	ø	ø	ø	+	+	+	ø	+	+
12	*cfxA*, *3*, *4*, *5*	0	*tet32*, *tetQ*	0	ø	ø	ø	ø	ø	ø	ø	+	+
13	0	*ermB*	*tetM*, *tetO*	0	ø	ø	ø	+	+	+	+	+	+
14	0	0	*tetM*	0	ø	ø	ø	+	+	+	ø	+	+
15	0	0	*tetQ*	0	ø	ø	ø	ø	ø	ø	ø	+	+
16	*cfxA*, *3*, *4*, *5*	0	0	0	ø	ø	ø	ø	ø	ø	ø	+	+
17	0	0	0	0	ø	ø	ø	+	ø	ø	ø	+	ø
18	0	0	0	0	ø	ø	ø	+	+	+	ø	+	+
19	0	0	0	0	ø	ø	ø	ø	ø	ø	ø	+	+

S, Susceptible (standard dosing regimen). R, Resistant. +, Microbiological growth of bacteria. ø, No microbiological growth of bacteria.

**Table 7 antibiotics-12-00068-t007:** Comparison of the different methods used for identification.

Sample	Resistance Genes	Identification via MALDI-TOF-MS/16S Sequencing	Identification via Shotgun Analysis	Results
1	*ermF*	*Prevotella intermedia*	*Prevotella intermedia*	same species
2	*ermF*	*Prevotella buccae*	*Prevotella nigrescens*	same genus
5	*ermF*	*Bacteroides thetaiotaomicron*, *Bacteroides ovatus*	*Bacteroides ovatus*	same species
7	*mefA*	*Gemella morbillorum*, *Anaplasma phagocytophilum*	*Gemella sanguinis*	same genus
8	*ermF*	*Prevotella maculosa*, *Corynebacterium durum*	*Prevotella melaninogenica*	same genus

## Data Availability

Not applicable.

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
