# Peer review of "Relationship between Phenotypic and Genotypic Resistance of Subgingival Biofilm Samples in Patients with Periodontitis"

_antibiotics, 2022, doi:10.3390/antibiotics12010068_

Round 1

Reviewer 1 Report

Thank you for allowing me to review this manuscript, whose purpose was to determine the relationship between the phenotypic and genotypic resistance against ampicillin-sulbactam, clindamycin, doxycycline and metronidazole of subgingival biofilm samples in periodontitis patients, using conventional methods.

There are items that authors need to refine before publication.

1. Write a better conclusion in the abstract.

2. Remove table 1 from the methods and transfer it to the results.

3. State when and where the research was conducted and the inclusion and exclusion criteria.

4. How was the sample size calculated?

5. Who are the researchers, and are they validated?

6. State the statistical package and methods? For example, has a correlation been made between the degree of periodontitis and other factors and the phenotypic and genotypic resistance of subgingival biofilm samples?

7. Better define the limiting factors and emphasize the study's strengths.

8. Should the conclusion be more specific and less general?

9. Did the study have a hypothesis, and was it confirmed?

Author Response

Dear editor,

We appreciate having the opportunity to resubmit our manuscript, entitled “Relationship between Phenotypic and Genotypic Resistance of Subgingival Biofilm Samples in Patients with Periodontitis” for consideration as an Original Article for publication in antibiotics. We are greatly appreciative of your kind suggestions, which have enabled us to improve our manuscript. Based on the comments we have improved the manuscript. Our point-by-point responses to each comment can be seen in the additional document. Our revisions based on your suggestions are marked up using the “Track Changes” function in the manuscript.

Reviewer 2 Report

In their manuscript entitledRelationship between Phenotypic and Genotypic Resistance of Subgingival Biofilm Samples in Patients with PeriodontitisSparbrod et al.  aimed to correlate the phenotypic resistance of bacteria in subgingival samples with the repertoire of resistance genes (genotype) and vice versa.

 General comment:

 This is an interesting and very well performed and written study.

 Some details for further improvement/consideration:

 Materials and Methods:

Lines 74ff: “CAL is in the case of periodontitis the distance between the cemento-enamel-junction, measured in mm” You mean “CAL is in the case of periodontitis the distance from the gingival margin to the cemento-enamel-junction, measured in mm”?!

Lines 96ff (Microbiological growth): Why not applying a control plate without any antibiotic? This limitation has a number of consequences listed below.

 Page 4, lines 125ff: while you are providing most of the standard protocols in extensive (or exhaustive) detail, you missed to provide amplification primer sequences and V-target region of the 16S-gene.

Lines 142ff: “The isolates had a minimum of 99% identity, except for one isolate of the Bacteroides ovatus species and one isolate of the Anaplasma ovis species. These two had a 96 % identity”. If it is below 97% you better write “like-species” in both cases.

Page 5, lines 185ff (Predictive values): “false-negative” (no gene but phenotypic resistance) could also be a result but you are not mentioning here. This will mainly occur on the MTZ-plate. Please include at least a note.

Results:

Page 6, lines 203ff: “a total of 63 [resistant] bacterial strains were isolated from the 19 subgingival biofilm samples”. Nice information, but how many are these in comparison to all cultivable strains? Again, the application of non-selective agar seems to be missing and a limitation of this study. Were resistant strains equally distributed or accumulation in a few subjects? Why were the S. mitis/oralis strains not be identified more exactly? This criticism is related to the missing information about primers used, especially as some V-regions are able to distinguish between these Viridans streptococci.

Lines 234ff: “All the 15 samples without resistance genes for ampicillin-sulbactam were phenotypic susceptible”, I guess “phenotypic susceptible” means no growths on the selective antibiotic-containing agar plate, but this is a kind of vague without a control without antibiotics. How can you be sure that the samples were preserved good enough for cultivation? Innsufficient preservation would lead to the same "no growth" result.

Page 6, lines 253ff: “We only illustrated these outcomes that were equal on the genus or species level.” I like your approach for matching pheno- and genotype. For this reason, I (and the reader in general, I guess) are interested in the overall results; please provide them as supplementary materials.

A comment on sample No. 5: Fecal Bacteroides were detected here; this is plausible and associated solely with this sample/patient; however, do you have an explanation? Was a contamination as source excluded?

Table 5: There should always be MTZ-resistant (thus facultative anaerobic) species in a subgingival sample; how do you explain “no growth” on in total four MTZ plates then?

If I do count all “+” of Table 5 together, I tell 71; in addition, on a few (e.g. MTZ-) plates you might have several different species, I guess. Please explain the discrepancy between 71 and the 63 actually isolated strains. Did you per se exclude the strains isolated from 16 samples plated on metronidazole due to the constitutive resistance here? Please clarify.

 A general comment: this is a well-conducted study, but the report of results is a kind of incomplete. Data on strains grown from MTZ-plates seem to be missing. Please provide a list of your isolates in correspondence to the sample number and sort of agar plate (e.g. as Suppl.). Other limits are the lack of a control agar plate and missing data about primers used for amplification and sequencing.

Author Response

(The authors gave the same response as above.)

Round 2

Reviewer 1 Report

Thanks author for accepted comments. Congratulations. 

Reviewer 2 Report

no further comments; well done